

# CAMELS-DE: hydro-meteorological time series and attributes for 1555 catchments in Germany

Ralf Loritz*[1], Alexander Dolich*[1], Eduardo Acuña Espinoza[1,A], Pia Ebeling[2,A], Björn Guse[3,4,A], Jonas Götte[5,6,7,A], Sibylle K. Hassler[8,A], Corina Hauffe[9,A], Ingo Heidbüchel[2,10,A], Jens Kiesel[3,11,A], Mirko Mälicke[1,A], Hannes Müller-Thomy[12,A], Michael Stölzle[13,A], Larisa Tarasova[14,A]

*equal contribution, A alphabetic order

[1]Karlsruhe Institute of Technology (KIT), Institute for Water and Environment, Karlsruhe, Germany

[2]Helmholtz Centre for Environmental Research - UFZ, Department Hydrogeology, Leipzig, Germany

[3]Kiel University, Hydrology and Water Resources Management, Kiel, Germany

[4]German Research Centre for Geosciences - GFZ Potsdam, Section Hydrology, Potsdam, Germany

[5]WSL Institute for Snow and Avalanche Research SLF, Davos Dorf, Switzerland

[6]Climate Change, Extremes and Natural Hazards in Alpine Regions Research Center CERC, Davos Dorf

[7]Institute for Atmospheric and Climate Science, ETH Zurich, Zurich, Switzerland

[8]Karlsruhe Institute of Technology (KIT), Institute of Meteorology and Climate Research - Atmospheric Trace Gases and Remote Sensing (IMK-ASF), Karlsruhe, Germany

[9]University of Technology Dresden (TUD), Institute of Hydrology and Meteorology, Dresden, Germany

[10]Bayreuth Centre of Ecology and Environmental Research, University of Bayreuth, Bayreuth, Germany

[11]Stone Environmental, 535 Stone Cutters Way, 05602 Montpelier (VT), USA

[12]Technische Universität Braunschweig, Leichtweiß-Institute for Hydraulic Engineering and Water Resources, Division of Hydrology and River Basin Management, Braunschweig, Germany

[13]Chair of Hydrology, University of Freiburg, Freiburg, Germany, now at: LUBW Landesanstalt für Umwelt (State Agency for Environment), Karlsruhe, Germany

[14]Helmholtz Centre for Environmental Research - UFZ, Department Catchment Hydrology, Germany

*Correspondence to*: Ralf Loritz (Ralf.Loritz@kit.edu) and Alexander Dolich (Alexander.Dolich@kit.edu)

**Abstract.** Comprehensive large sample hydrological datasets, particularly the CAMELS datasets (Catchment Attributes and Meteorology for Large-sample Studies), have advanced hydrological research and education in recent years. These datasets integrate extensive hydrometeorological observations with landscape features, such as geology and land use, across numerous catchments within a national framework. They provide harmonised large sample data for various purposes, such as assessing the impacts of climate change or testing hydrological models on a large number of catchments. Furthermore, these



datasets are essential for the rapid progress of data-driven models in hydrology in recent years. Despite Germany's extensive
hydrometeorological measurement infrastructure, it has lacked a consistent, nationwide hydrological dataset, largely due to
its decentralised management across different federal states. This fragmentation has hindered cross-state studies and made
the preparation of hydrological data labour-intensive. The introduction of CAMELS-DE represents a step forward in
bridging this gap. CAMELS-DE includes 1555 streamflow gauges with hydro-meteorological time series data covering up to
70 years (median length of 46 years and a minimum length of 10 years), from January 1951 to December 2020. It includes
consistent catchment boundaries with areas ranging from 5 to 15,000 km$^2$ along with detailed catchment attributes covering
soil, land cover, hydrogeologic properties and data about human influences. Furthermore, it includes a regionally trained
Long-Short Term Memory (LSTM) network and a locally trained conceptual model that were used as quality control and that
can be used to fill gaps in discharge data or act as baseline models for the development and testing of new hydrological
models. Given the large number of catchments, including numerous relatively small ones (617 catchments < 100 km$^2$), and
the time series length of up to 70 years (156 catchments), CAMELS-DE is one of the most comprehensive national
CAMELS datasets available and offers new opportunities for research, particularly in studying long-term trends, runoff
formation in small catchments and in analysing catchments with strong human influences.

## 1 Introduction

The CAMELS (Catchment Attributes and MEteorology for Large-sample Studies) datasets have become a cornerstone
within the hydrological community for their comprehensive and consistent integration of hydro- and meteorological data
across entire countries, including the USA, UK, Australia, Brazil, Chile, and others (e.g. Addor et al., 2017, Coxon et al.,
2020). These datasets combine catchment attributes (e.g. land use, geology, and soil properties), hydrological time series
(e.g. water level and discharge), and meteorological time series (e.g. precipitation and temperature) for a multitude of
catchments typically within a single country. A distinctive feature of CAMELS datasets is their role as a benchmark for
hydrological modelling and large sample analysis, enabling the comparison of hydrological models and the validation of
water resources management strategies across diverse landscapes and climates (Brunner et al., 2021). Particularly the
CAMELS-US dataset has thereby formed the basis for the on-going rise of machine learning methods in hydrology (e.g
Kratzert et al., 2019).

Despite the widespread adoption and utility of CAMELS datasets in research, teaching, and practical applications globally,
Germany with its extensive hydro-meteorological measurement network has no comprehensive and harmonised dataset yet.
While there are large sample hydrological datasets that cover either parts of Germany (Klingler et al., 2021), only a fraction
of the available national hydrological data (Färber et al., 2023), or focus on catchment water quality and thus cover a lower
sampling frequency (Ebeling et al., 2022), the absence of a full CAMELS dataset that includes harmonised, daily,
high-quality national hydrological and meteorological data together with catchment attributes and catchment boundaries





derived from national and international products limits the potential for comprehensive analyses and advancements in
hydrological research and practice. The CAMELS-DE data set addresses this gap (Dolich et al., 2024). CAMELS-DE
compiles discharge, water levels, catchment attributes, and catchment boundaries together with a suite of meteorological
time series and catchment attributes for 1555 catchments across Germany. Furthermore, it provides discharge simulations
both from a regional trained Long-Short Term Memory (LSTM) network and a local conceptual hydrological model with the
dataset that can act as a benchmark for future modelling studies in Germany or be used to fill missing data gaps in the
hydrological time series. Each component of the CAMELS-DE processing pipeline is fully containerized (see section 7),
which solves code dependency issues and generally contributes to the traceability, comprehensiveness, and reproducibility of
the generation of CAMELS-DE. This study introduces not only a comprehensive dataset but also a suite of tools designed to
generate reproducible hydrological datasets from the provided raw data. In the following sections we provide a
comprehensive description of all data contained within CAMELS-DE including (1) its source data, (2) how the time series
and attributes were produced, and (3) a discussion of the associated limitations and uncertainties. The structure of this paper
(and also the corresponding dataset) closely mirrors that of the CAMELS-UK (Coxon et al., 2020) and CAMELS-CH (Höge
et al., 2023) studies, ensuring comparability of the datasets while maintaining distinct elements that are not identical but
closely related.
**2 Data sources and providers**
CAMELS-DE brings together hydrological data, consisting of daily measurements of discharge (m³/s) and water levels (m),
from twelf German federal state agencies, namely the Landesanstalt für Umwelt Baden-Württemberg (LUBW, Nomenclature
of Territorial Units for Statistics (NUTS) Level 1: DE1), Bayerisches Landesamt für Umwelt (LfU-Bayern, DE2),
Landesamt für Umwelt Brandenburg (LfU-Brandenburg, DE4), Hessisches Landesamt für Naturschutz, Umwelt und
Geologie (HLNUG, DE7), Landesamt für Umwelt, Naturschutz und Geologie Mecklenburg-Vorpommern (LUNG MV,
DE8), Niedersächsischer Landesbetrieb für Wasserwirtschaft, Küsten- und Naturschutz, Landesamt für Natur (NLWKN,
DE9), Umwelt und Verbraucherschutz Nordrhein-Westfalen (LANUV NRW, DEA), Landesamt für Umwelt Rheinland-Pfalz
(LUA-Rheinland Pfalz, DEB), Landesamt für Umwelt, Landwirtschaft und Geologie Sachsen (LfULG, DED), Landesamt
für Umweltschutz Sachsen-Anhalt (LAU, DEE), Landesamt für Landwirtschaft, Umwelt und ländliche Räume
Schleswig-Holstein (LLUR, DEF), and Thüringer Landesamt für Umwelt, Bergbau und Naturschutz (TLUBN, DEG). The
only federal states not included are the city-states of Bremen, Hamburg, and Berlin, along with the federal state Saarland,
which together account for less than 1.5 % of Germany's area, ensuring that the CAMELS-DE dataset remains representative
for Germany.
Meteorological data, specifically precipitation, temperature, relative humidity and radiation, were obtained from the German
Weather Service (DWD) from the HYRAS dataset (DWD-HYRAS, 2024). Spatially aggregated catchment attributes were





obtained from various sources. From the European Union, we incorporated open-access datasets from Copernicus, the EU's
Earth observation program, in particular the Copernicus GLO-30 DEM (Global 30-meter Digital Elevation Model;
EU-DEM, 2022) for information about topography and the CORINE Land Cover 2018 dataset (CLC, 2018) for information
about land cover. Soil attributes were derived from the global SoilGrids250m dataset (Poggio et al., 2021). Hydrogeological
catchment attributes were derived from the "Hydrogeologische Übersichtskarte von Deutschland 1:250.000" (HGM250,
2019) provided by the Bundesanstalt für Geowissenschaften und Rohstoffe (BGR) while information about human
influences, e.g. dams or weirs, was sourced from Speckhann et al. (2021).

## 103  3 Catchments

For CAMELS-DE, we sourced discharge ($m^3 s^{-1}$), water level data (m) and metadata for 2964 gauges and water level stations
from the different federal state agencies (see section 2). We created a subset of the data by selecting only measurement
stations that contained all required information, such as gauge name, location and catchment area in their metadata (n = 2700
stations), have at least a total of 10 years of discharge data, which must not necessarily be continuous (n = 2227 stations), are
larger than 5 km² and smaller than 15,000 km² (n = 2586 stations), are located entirely within the borders of Germany (n =
2298 stations) and where the derived catchment area does not differ more than 20 % from the reported value by the federal
states (n = 2164 stations; see section 3.1). These requirements were based on the following reasoning: At least 10 years of
discharge data are required to ensure that a sufficient time series length is available to perform hydrological modelling and
calculate hydrological signatures. These requirements were established based on the following rationale: A minimum of 10
years of discharge data is necessary to ensure an adequate time series length for hydrological modeling and calculating
hydrological signatures. The minimum catchment area of 5 km² was selected because some meteorological raster products
have a resolution of 5 x 5 km. The upper limit was set because catchments larger than 15,000 km² are predominantly
influenced by human activities and often extend beyond Germany's borders, necessitating their exclusion. The 20 %
discrepancy between derived and reported catchment areas was arbitrarily chosen as an acceptable threshold for mass
balance errors. This threshold prevents the inclusion of catchments with significantly inaccurate delineations while avoiding
the exclusion of too much data (see Fig. 2b). Catchments partially located outside Germany's borders were excluded to avoid
complications with cross-border data, especially given the absence of open, high-quality meteorological data from the DWD
beyond Germany's national borders from 1951 to 2020. These criteria resulted in a subset of 1555 gauges for the
CAMELS-DE dataset, which provides a reliable representation of hydrological processes in Germany (Fig. 1c, d).

### 123  3.1 Catchment boundaries

Not all state authorities provided official catchment boundaries for their gauging stations, and the methods used by the
federal states to derive these boundaries are not uniform and remain unclear. Therefore, we tested two different global
catchment datasets, HydroSHEDS (Lehner et al., 2021) and MERIT Hydro (Yamazaki et al., 2019), to derive a consistent set





of catchment boundaries across Germany for the CAMELS-DE dataset. For that we compared the catchment areas
determined with HydroSHEDS and MERIT Hydro to the catchment areas reported by the state authorities. This comparison
was possible because all federal states shared the area of the catchments while not always sharing the actual catchment
boundaries. Overall, the comparison revealed that MERIT Hydro has lower errors between the reported and derived
catchment areas compared to HydroSHEDS. Among other reasons, this is because MERIT Hydro derives the catchment
boundaries directly at the gauge locations provided by the federal states (see section 3.2). The comparison between MERIT
Hydro and HydroSHEDS was further supported by extensive manual assessments, involving the visual inspection of
numerous catchments to evaluate their shapes and alignments in case the federal state provided the data. Consequently,
MERIT Hydro was used for the derivation of catchment boundaries for CAMELS-DE. Note that the derivation of the
catchment boundaries is a major source of uncertainty as the meteorological time series and the catchment attributes are
dependent on the catchment boundaries. To minimise the uncertainty of the catchment delineation we only included
catchments with a deviation of up to 20 percent from the catchment area reported by the federal agencies (Fig. 2b). We report
the original catchment area as (area_metadata) and the MERIT-Hydro based area (area) in the table of topographic attributes
(Table 2).

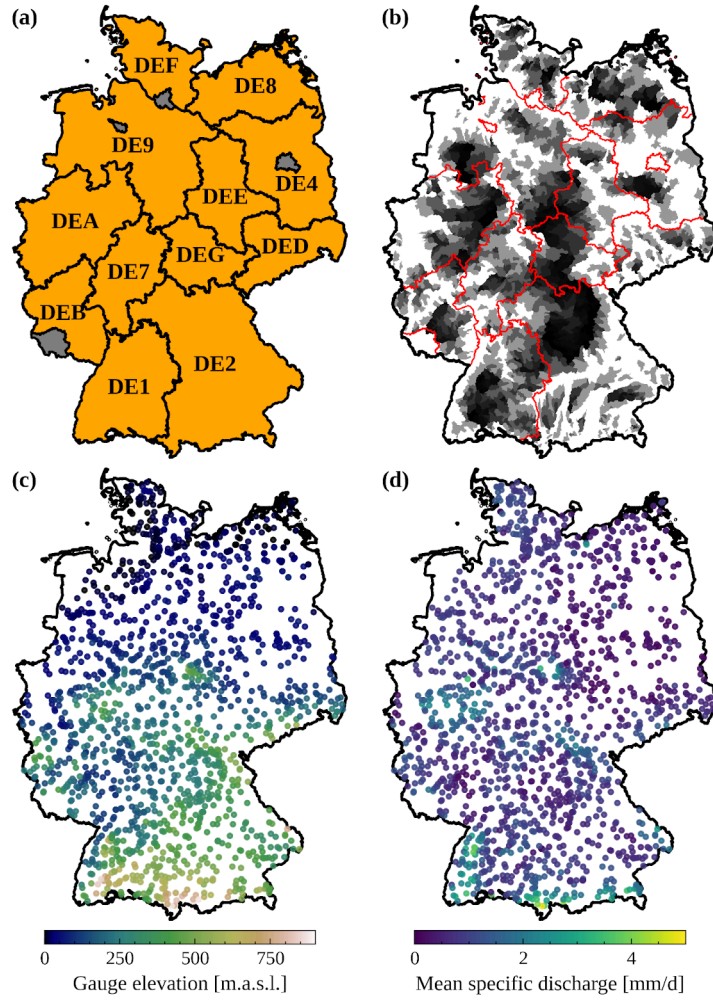

**Figure 1:** Panel (a) shows the German federal states labelled with their NUTS Level 1 ID as used for the CAMELS-DE gauge IDs. Panel (b) shows all 1555 catchments provided in CAMELS-DE, the geometries of the catchments are shown transparently, so a darker colour means that the geometries of the catchments in that area overlap; the darker the colour, the higher the density of catchments in that area. Panel (c) and panel (d) show the location of all 1555 gauging stations in CAMELS-DE; in panel (c) the locations are coloured according to the elevation of the gauging station, while in panel (d) the locations are coloured according to their mean specific discharge value. borders of Germany: © GeoBasis-DE / BKG (VG250, 2023)

## 3.2 Catchment boundaries derived from MERIT Hydro

MERIT (Multi-Error-Removed Improved-Terrain) Hydro was released by Yamazaki et al. (2019); providing a global hydrography dataset based on the MERIT DEM and various maps of water bodies (e.g. Global 3 arc-second Water Body Map by Yamazaki et al., 2017). It includes information such as flow direction, flow accumulation, adjusted elevations for hydrological purposes, and the width of river channels. The delineator.py package (Heberger, 2023) was used to delineate catchment boundaries. The method automatically derives catchment boundaries from the MERIT Hydro dataset based on the longitude and latitude of a gauging station and snaps the catchment pour point to the closest stream. Fig. 1b. shows all

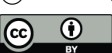



derived catchments using MERIT Hydro within the German borders. The median catchment area within CAMELS-DE is
131.3 km² (Fig. 2a). Compared to other CAMELS datasets, CAMELS-DE includes a large number of relatively small
catchments with an area of less than 100 km² (i.e. 617 catchments). Uncertainties in catchment delineation arise when
comparing areas reported by federal states with those derived from MERIT Hydro, as shown in Fig. 2b, and these
discrepancies are not uniformly distributed across Germany. They tend to be higher in regions with minimal topography,
particularly in the federal states to the north and east of Germany. Consequently, a large number of catchments are excluded
from the CAMELS-DE dataset in the northern parts of Germany due to mismatches between reported and estimated areas. In
the federal states of Brandenburg (DE4) and Mecklenburg-Western Pomerania (DE8), for example, we received 447 gauging
stations, but given the uncertainty of the delineation in flat areas, only 277 of them showed a deviation of less than 20
percent from the reported area. In contrast, in the more mountainous state of Baden-Württemberg (DE1), 225 of 241
catchments met this criterion. As we report both the catchment areas provided by the federal states and those estimated by
MERIT Hydro, the differences between these two measurements can be used to select or exclude catchments where there are
significant uncertainties in the catchment shape and correspondingly in the derived static and dynamic attributes.

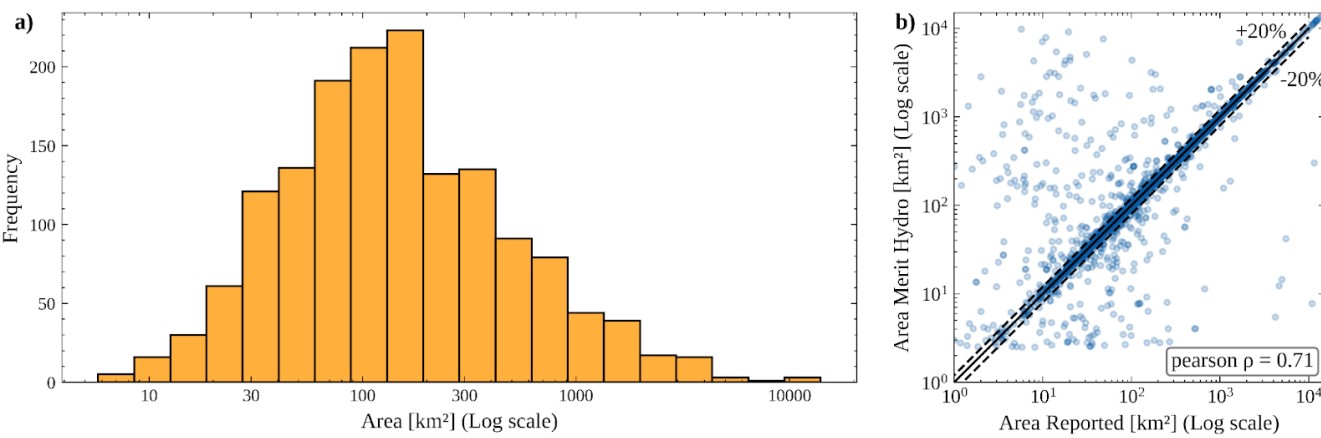


**Figure. 2:** Panel (a) shows the distribution of catchment areas on a logarithmic scale. Panel (b) shows the accuracy of catchment areas derived using
MERIT Hydro compared to the area reported by the federal agencies; the dashed lines indicate ±20 percent error tolerance that was set for catchment
selection.

## 171 4 Time series

CAMELS-DE comprises an observation-based and a simulation-based set of hydro-meteorological daily time series as
specified in Tab. 1 for a period from 1 January 1951 to 31 December 2020. Note that we do not include any information on
evaporation in the non-simulated time series data, as we only include observation-based data here. However, a time series of
potential evaporation based on the temperature-based Hargreaves methodology is included in the simulated data (see section
6.2 for more details). However, due to the simplicity of the chosen approach, the potential evapotranspiration time series are



highly uncertain, and one should exercise caution when using them.

All meteorological forcing data within CAMELS-DE are sourced from the Hyras datasets, which are based on the
interpolation of meteorological station data (DWD-HYRAS, 2024). The reliability of these datasets can be compromised by
the individual interpolation methods employed (see section 4.1 to 4.3). In addition, inaccuracies in meteorological
measurements can introduce uncertainties in the generated grid fields, especially given the extended timescale of 70 years,
which may include changes in location and sensor types. Another source of uncertainty is the fact that the number of stations
used in the interpolation process varies over time, mirroring changes in the measurement network. For example, the number
of stations used for interpolating precipitation data fluctuates, starting at around 4500 in 1951, peaking at approximately
7500 in 2000, and then decreasing to approximately 5000 by 2020. In contrast, the number of stations used for radiation
interpolation shows a consistent increase over the years, though the total number remains significantly lower, reaching about
900 stations by 2020. This uncertainty is crucial to consider when comparing data across different years, particularly if the
focus is on a single or a few catchments in a certain area. Finally, we use the 'exact extract' method, which ensures that
raster cells that are only partially covered are treated properly as they are weighted by the proportion of the cell that is
covered, i.e. a raster cell that is only 20 % covered by the catchment is only weighted by 20 % when we aggregate to the
spatial catchment mean. This is particularly important when deriving meteorological data for very small catchment areas.
Although this approach also aids in comparing products with different resolutions, it is important to consider that the spatial
resolution of the precipitation data, at 1 x 1 km, offers finer detail compared to the 5 x 5 km resolution used for temperature,
humidity, and radiation data. This difference is crucial when comparing these datasets within smaller catchments.

## 4.1 Precipitation

CAMELS-DE utilises precipitation data (mm d$^{-1}$) with daily resolution, sourced from the HYRAS-DE-PRE dataset v5.0
(HYRAS-DE-PRE, 2022). We have calculated daily minimum, mean, median, maximum, and standard deviation of the
rainfall field over the catchment. We estimated these statistical measures, rather than just the mean, because this allows us to
capture variations and patterns that can be crucial for event characterization or rainfall-runoff modelling, as illustrated in Fig.
3. The HYRAS-DE-PRE dataset v5.0 dataset is produced using the REGNIE interpolation method (Rauthe et al., 2013),
which employs daily measured values from meteorological stations to generate an interpolated product on a 1x1 km grid. A
detailed description of the interpolation method and the related uncertainties can be found in the official data description
(HYRAS-DE-PRE, 2022).



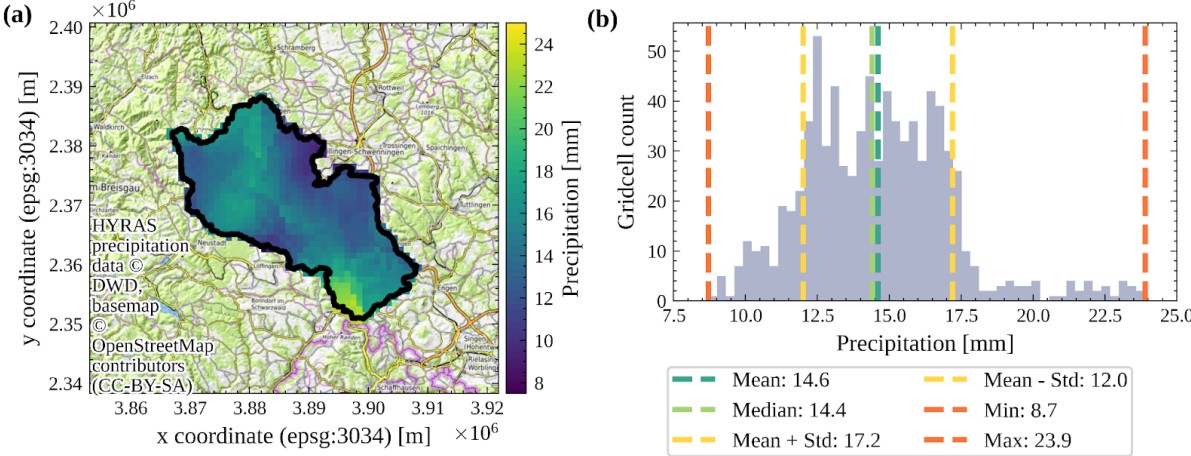

**Figure 3:** Panel (a) shows the catchment boundaries (black line) of the catchment Kirchen-Hausen in Baden-Württemberg overlayed by a clipped daily precipitation field from the HYRAS dataset on the date 1951-02-20. Panel (b) shows the spatial distribution of rainfall during the same high precipitation event as (a) over the catchment on 1951-02-20 and the statistical moments (mean, median, standard deviation, minimum and maximum) derived from the spatial distribution.

## 4.2 Temperature and relative humidity

CAMELS-DE employs daily temperature (°C) and relative humidity (%), derived from the HYRAS-DE-TAS (HYRAS-DE-TAS, 2022), TASMIN (HYRAS-DE-TASMIN, 2022), TASMAX (HYRAS-DE-TASMAX, 2022), and HURS (HYRAS-DE-HURS, 2022) datasets v5.0, which cover the period from 1951 to 2020 on a 5 km x 5 km grid. This includes the mean, median, and standard deviation of temperature from HYRAS-DE-TAS, alongside the minimum and maximum temperatures from TASMIN and TASMAX, respectively. Additionally, for humidity, we integrate daily minimum, mean, median, maximum, and standard deviation values across the catchment area. The temperature and humidity data is based on interpolated station values (Razafimaharo et al., 2020). This interpolation method involves a nonlinear regression at each time step, aiming to estimate regional vertical temperature profiles across 13 subregions. These subregions are delineated based on criteria such as weather divides, proximity to the coast, and the extent of north-south variation. A detailed description of the interpolation method and the related uncertainties can be found in the corresponding data descriptions (HYRAS-DE-TAS, (2022); HYRAS-DE-TASMIN, (2022); HYRAS-DE-TASMAX, (2022); HYRAS-DE-HURS, (2022)).

## 4.3 Radiation

The CAMELS-DE dataset utilises daily mean global radiation data (in W m$^{-2}$) derived from the HYRAS-DE-RSDS datasets v3.0 (HYRAS-DE-RSDS, 2023), that covers a period from 1951 to 2020 with a 5 km x 5 km grid. We have derived daily minimum, mean, median, maximum, and standard deviation of the radiation field over the catchment. The global radiation (RSDS) dataset integrates station measurement data (including sunshine duration and global radiation), satellite data, and





ERA5 data (Muñoz-Sabater et al., 2021). A detailed description of the interpolation method and the related uncertainties can
be found in the official data description (HYRAS-DE-RSDS, 2023).

### 4.4 Discharge and water levels

Observed discharge and water level data were requested from 13 and delivered from 12 federal state agencies (see section 2)
as time series recorded at the gauging stations (Tab. 1). The number of stations with daily discharge data available per year
increases in time from 187 on 1 January 1951 to a maximum of 1459 between November 2010 and February 2011 (Fig. 4a).
The number of stations with water level data is generally lower, starting at 110 stations on 1 January 1951 and reaching a
maximum of 1444 stations between March 2015 and December 2015. The time series span a maximum of 70 years, with
each measuring station providing at least 10 years of data between January 1951 and December 2020 (Fig. 4b). These 10
years do not need to be consecutive but typically are. The median time series length of discharge is 46 years, while the
median time series length of water level is 40 years. There is a sharp drop-off in Fig. 4a of 137 stations without data from
2017 to 2018 as the provided data from NLWKN (Lower Saxony, DE9) only ranges until the end of 2017. Another anomaly
in Fig. 4a is the drop immediately followed by a rise in the year 2020, which is due to the fact that all measuring stations in
Rhineland-Palatinate (DEB) show a gap in the discharge data from 10 February 2020 to 15 February 2020 and in the water
level data from 13 February 2020 to 15 February 2020. No explanation could be found for this gap. The remaining data after
the gap was manually quality controlled by visual inspection of the observed and simulated time series and no reason to
exclude this data was found. In total, CAMELS-DE includes 156 stations for which the entire temporal range of 70 years of
discharge data is available and for which a maximum of 2 percent of the data is missing in this period. There are 85 stations
where this is the case for water level data. The quality control of all discharge and water level data was conducted by the
respective federal states (quality controlled data was requested). However, the specific methods employed in this quality
control are neither the same across the states, nor are they documented in some cases. Typically, quality control entails that a
technical clerk has visually inspected the hydrological time series data. To account for this uncertainty we conducted an
additional review of all time series data for high negative values and unrealistically high outliers and replaced such data
points with NaN values. We were conservative in these cases and only deleted values that were clear data errors to not
remove potential extreme flood events from the time series. This adjustment was necessary in 7 catchments and is
documented in the processing pipeline to assure reproducibility. Moreover, we assessed the hydro-meteorological time series
using both a hydrological model and a data-driven model. This analysis helped us identify catchments with weak correlations
between meteorological conditions and hydrological responses as well as catchments in which the mass balance is far from
being closed. All catchments that exhibited a low model performance of the conceptual model were subjected to manual
visual inspection, resulting in the removal of 14 catchments (for more details we refer to section 6).





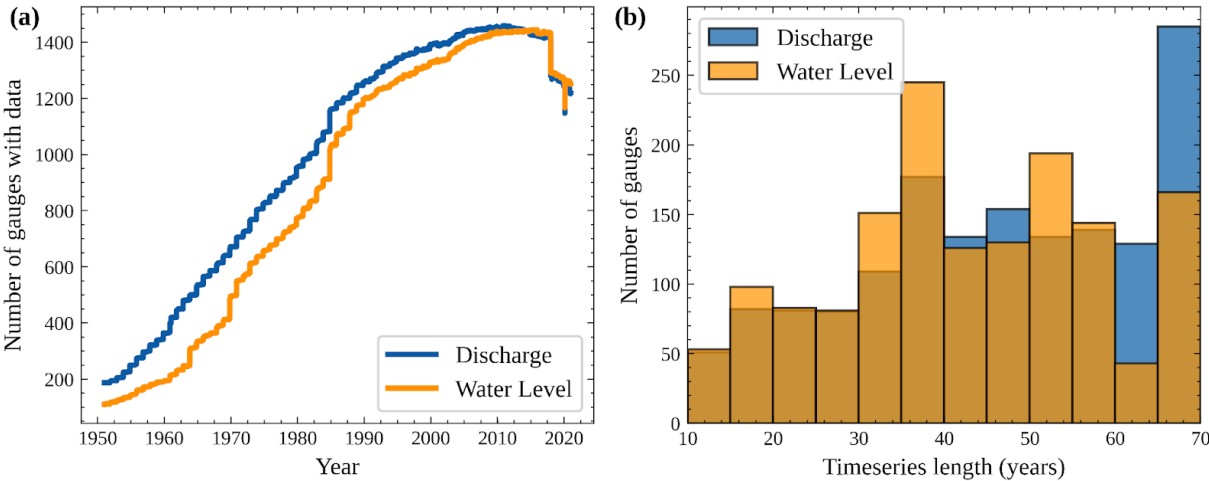

**Figure 4:** Panel (a) shows the number of gauging stations with available discharge (blue) and water level data (orange) in the period from 1951 to 2020, taking into account data gaps, i.e. the data must actually be available at the respective time. Panel (b) shows the time series length of the discharge and water level observations in CAMELS-DE. Possible gaps in the data are not taken into account in the time series length; in this case, the time series length is the number of years from the first available value to the last available value of a station.

**Table 1:** Catchment-specific hydro-meteorological variables available as daily time series in CAMELS-DE

| Time series class | Time series name | Description | Unit | Data source |
|---|---|---|---|---|
| Hydrologic time series (1 Jan 1951–31 Dec 2020) | discharge_vol | Catchment discharge calculated from the water level and gauge geometry | $m^3\ s^{-1}$ | Federal state agencies (see section 2) |
| | discharge_spec | Observed catchment-specific discharge (converted to millimetres per day using catchment areas described in section 3.1) | $mm\ d^{-1}$ | |
| | water_level | Observed daily water level | m | |
| Meteorologic time series (1 Jan 1951–31 Dec 2020) | precipitation_mean, precipitation_median, precipitation_min, precipitation_max, precipitation_std | Observed interpolated spatial mean, median, minimum, maximum and standard deviation of the daily precipitation (1x1 km$^2$) | $mm\ d^{-1}$ | German Weather Service HYRAS (DWD-HYRAS, 2024) |
| | temperature_min | Observed interpolated spatial mean daily minimum temperatures (5x5 km$^2$) | °C | |
| | temperature_mean | Observed interpolated spatial mean daily mean temperatures (5x5 km$^2$) | °C | |
| | temperature_max | Observed interpolated spatial mean daily maximum temperatures (5x5 km$^2$) | °C | |
| | humidity_mean, humidity_median, humidity_min, | Observed interpolated spatial mean, median, minimum, maximum and standard deviation of the daily humidity (5x5 km$^2$) | % | |



| | humidity_max, humidity_std | | | |
|---|---|---|---|---|
| | radiation_global, radiation_median, radiation_min, radiation_max, radiation_std | Observed interpolated spatial mean, median, minimum, maximum and standard deviation of the global radiation (5x5 km$^2$) | W m$^2$ | |
| Simulated hydrologic time series (1 Jan 1951–31 Dec 2020) | pet_hargreaves | Daily mean of potential evapotranspiration calculated using the Hargreaves equation | mm d$^{-1}$ | Regional LSTM model, conceptual model and Hargreaves equation for potential evapotranspiration (see section 6, https://github.com/KIT-HYD/Hy2DL/tree/v1.1, last access: 24 July 2024) |
| | discharge_vol_obs | Observed volumetric discharge | m$^3$ s$^{-1}$ | |
| | discharge_spec_obs | Observed catchment-specific discharge | mm d$^{-1}$ | |
| | discharge_vol_sim_lstm | Volumetric discharge calculated from discharge_spec_sim_lstm and the catchment area | m$^3$ s$^{-1}$ | |
| | discharge_spec_sim_lstm | Catchment-specific discharge simulated with the LSTM (see section 6) | mm d$^{-1}$ | |
| | discharge_vol_sim_conceptual | Volumetric discharge calculated from discharge_spec_sim_conceptual and the catchment area | m$^3$ s$^{-1}$ | |
| | discharge_spec_sim_conceptual | Catchment-specific discharge simulated with the conceptual model (see section 6) | mm d$^{-1}$ | |
| | simulation_period (training, validation, testing) | Flag indicating the simulation period in which the daily value is contained (training, validation, testing) | – | |

## 5 Catchment attributes

In addition to the daily time series of hydro-meteorological variables available in CAMELS-DE, the dataset also includes a series of static catchment attributes which are considered time-invariant and include information about topography (section 5.1), hydroclimatic signatures (section 5.2) and catchment attributes covering land-cover (section 5.3), soil (section 5.4), hydrogeology (section 5.5) and human influences (section 5.6).

### 5.1 Location and topography

For CAMELS-DE, we developed a system of catchment IDs, since the official IDs used by the federal states are inconsistent beyond federal state boundaries. However, the official provider IDs are contained in the topographic attributes of the dataset (Tab. 2). The gauge IDs in CAMELS-DE are based on the NUTS classification, which divides the EU territory hierarchically according to political boundaries. In Germany, the first hierarchical level NUTS 1 provides a code for each federal state (e.g. DE7 for Hessen, DED for Saxony; Fig. 1b). We assign an ID code to each gauge as follows. The ID of each gauge starts with the NUTS 1 code of the corresponding federal state. For each federal state the gauges are coded in arbitrary order starting from 10000 for the first gauge and adding a step of 10 for each following gauge (e.g. DE710000 for the first station in



Hessen, DE710010 for the second station, DE710020 for the third station, etc.). This system ensures consistency of the
gauge IDs in Germany, and additionally immediately provides the information about the federal state of each gauge.
Topographic attributes such as the location (coordinate systems WGS84 and ETRS89), gauge elevation (m) and catchment
area ($km^2$) were provided by the federal agencies, the area of the MERIT Hydro catchment is also provided. Additionally we
derived the gauge point elevation (m) and basic statistical variables (min, mean, median, 5th and 95th percentile, max) of the
catchment elevation (m) from the GLO-30 DEM. CAMELS-DE additionally provides the location of all gauging stations and
catchment boundaries as a shape file and a geopackage file.

## 284 5.2 Climate and hydrology

For the CAMELS-DE dataset, we calculated long-term climatic and hydrological signatures in line with the attributes found
in CAMELS-CH (covering the period between 1981–2020) and CAMELS-UK (covering the period between 1970–2015)
with the difference that we cover the period from 1951–2021 (see Tab. 2). Both types of attributes are calculated based solely
on complete hydrological years with respect to the discharge (1 October to 30 September of the following year; again inline
with the definition of a hydrological year chosen in CAMELS-UK and CAMELS-CH), with a maximum tolerance of 5 %
missing values per hydrological year, ensuring robustness in the data used for analysis. If a specific catchment has discharge
data for only a limited number of hydrologic years, we calculate the climatic and hydrological indices for those same years to
maintain consistency across all CAMELS datasets and across the climatic and hydrological attributes.

For each catchment, the hydrologic attributes include values for the mean specific discharge (mm $d^{-1}$), the runoff ratio, the
start and end dates of available discharge data, the percentage of days on which discharge data is available (%), the slope of
the flow duration curve between the log-transformed 33rd and 66th percentiles, the day on which the cumulative discharge
since 1 October reaches half of the annual discharge (d), the 5th and 95th quantile of specific discharge (mm $d^{-1}$) and the
frequency of high flow, low flow and zero flow days (d $yr^{-1}$) together with the average duration of high-flow and low-flow
events (d). The climatic attributes are calculated on the basis of the HYRAS meteorological data for each catchment and
include mean daily precipitation (mm $d^{-1}$), the seasonality of precipitation, the fraction of precipitation falling as snow, the
frequency of high and low precipitation days (d $yr^{-1}$), the average duration of high precipitation events and dry periods (d) as
well as the season during which most high and low precipitation days occur. The code to estimate the signatures in
CAMELS-DE is based on the codes used to derive the signatures for CAMELS-US (https://github.com/naddor/camels, last
access: 19 July 2024), CAMELS-UK and CAMELS-CH to assure compatibility.

## 305 5.3 Land cover

Land cover in CAMELS-DE is derived from the Corine Land Cover dataset (CLC, 2018) which provides consistent and
thematically detailed information on land cover across Europe. The dataset was produced within the frame of the Copernicus
Land Monitoring Service referring to land cover / land use status of the year 2018 and is based on the classification of



satellite images. CAMELS-DE includes land cover percentages per catchment of the first hierarchical land cover level: artificial surfaces, agricultural areas, forests and semi-natural areas, wetlands and water bodies. The decision to not mix the hierarchical land cover levels ensures that uncertainties in classification due to varying levels of detail are minimised. Catchment shapes and codes to derive land cover classes of lower order in a consistent manner with CAMELS-DE are delivered with the dataset (Dolich, 2024).

## 5.4 Soil

Soil attributes for CAMELS-DE are derived from the SoilGrids250m dataset (Poggio et al., 2021), which maps the spatial distribution of soil properties globally at six standard depths. The SoilGrids dataset is generated by training a machine learning model on approximately 240,000 locations worldwide, using over 400 global environmental covariates that describe vegetation, terrain morphology, climate, geology, and hydrology. For CAMELS-DE, we derived the mean values of the soil bulk density, soil organic carbon, volumetric percentage of coarse fragments and proportions of clay, silt and sand for each catchment. The resulting variables are aggregated from the six SoilGrid depths to the depths 0-30 cm, 30-100 cm and 100-200 cm by calculating a weighted mean. The accuracy of soil property models, as described by Poggio et al. (2021), is limited by the availability and quality of input data and the assumptions in the modelling process. For instance, discrepancies in how soil data are collected, analysed, and reported by different entities challenge efforts toward data standardisation and harmonisation. However, the relatively high number of observations in Germany reduces this uncertainty to a certain extent. Furthermore, the defined catchment boundaries allow for an assessment of the reported uncertainties within each catchment. If needed the catchment boundaries delivered with CAMELS-DE can be used to calculate the reported uncertainties of SoilGrids within each catchment.

## 5.5 Hydrogeology

The hydrogeological attributes for CAMELS-DE are derived from the hydrogeological overview map of Germany on the scale of 1:250,000; "HÜK250" (HGM250, 2019), which describes the hydrogeological characteristics of the upper, large-scale contiguous aquifers in Germany. For CAMELS-DE, the areal percentage of the various HÜK250 classes (see Tab. 2) was calculated for each catchment, whereby the variables of the classes permeability, aquifer media type, cavity type, consolidation, rock type and geochemical rock type sum to 100 percent. Uncertainties in these data may arise from the generalisation required to scale point measurements to a gridded product, which can oversimplify complex hydrogeological features, potentially leading to inaccuracies in the representation of local variations and the spatial distribution of aquifer properties.

## 5.6 Human influence

CAMELS-DE includes information on human influences within catchments, primarily focusing on existing dams and reservoirs. This information is sourced from the inventory of dams in germany (Speckhann et al., 2021), which offers





detailed data including dam names, locations, associated rivers, years of construction and operation start, crest lengths, dam
heights, lake areas, lake volumes, purposes (such as flood control or water supply), dam structure types, and specific building
characteristics for 530 dams across Germany. For catchments containing multiple dams, this data is aggregated to provide a
comprehensive overview. Specifically, CAMELS-DE includes key information about the dams within each catchment, such
as the number of dams, the names of the dams, the rivers where these dams are located, the operational years of the oldest
and newest dams, the total area and volume of all dam lakes at full capacity, and the overall purposes of these dams. It is
important to note that the "Inventory of Dams in Germany" does not claim to be exhaustive. The absence of recorded dams
in this inventory does not necessarily indicate a lack of human influence within a catchment. Nearly all catchments in
Germany experience substantial anthropogenic influences, and it is likely that some dams, weirs, or reservoirs (particularly
smaller ones) are not documented in the dataset. Another relevant indicator of human influence included in CAMELS-DE is
hence the proportion of artificial and agricultural surfaces derived from land cover attributes (see section 5.3).

## 6 Benchmark LSTMs and conceptual models

CAMELS-DE, in addition to hydro-meteorological observations and catchment attributes, includes results from data-driven
and conceptual lumped rainfall-runoff simulations for each catchment. More specifically, these results are derived from a
regionally trained LSTM network (trained on all catchments at the same time) and a locally trained lumped conceptual
hydrological model (trained at each individual catchment). These models serve three main purposes: (a) they are used to
identify catchments where the relationship between meteorological forcing and streamflow is difficult to capture (low model
performance), indicating possible strong human influences such as dams or reservoirs, or potential issues with the catchment
delineation or the streamflow or meteorological time series; (b) they can serve as a benchmark for future modelling studies
based on CAMELS-DE in a sense that the reported performance values and time series can be used as a baseline model and
(c) in case of a good model performance can be used to fill missing values of the observed discharge time series. Both
models were trained over the period from October 1, 1970, to December 31, 1999, validated from October 1, 1965, to
September 30, 1970, and tested from January 1, 2000, to December 31, 2020. CAMELS-DE includes the simulated
discharges for both models for the entire 70 years (Tab. 1), a flag was added to indicate if the corresponding time step was
used in training, validation or testing. In the following we explain the model setups and analyse the simulation results in
detail. The code of the LSTM model and the conceptual hydrological model were carefully tested and benchmarked (Acuña
Espinoza et al., 2024). The codes have been designed to allow easy access and a permalink to the code version used for
CAMELS-DE can be found here (https://github.com/KIT-HYD/Hy2DL/tree/v1.1, last access: 24 July 2024).

### 6.1 Setup LSTM model

The LSTM uses mean precipitation, standard deviation of precipitation, mean radiation, mean minimum temperature and
mean maximum temperature as dynamic (time varying) input features and specific discharge as a target variable. Static



features and hyperparameters were set according to the study of Acuña et al. (2024) with modifications made to (1) an
increased hidden size from 64 to 128 and (2) a reduced number of epochs from 30 to 20. The remaining hyperparameters
were set as follows: number of hidden layers = 1; learning rate = 0.001; dropout rate = 0.4; batch size = 256; sequence length
= 365 days; iterative optimization algorithm = Adam. We use the basin-averaged Nash-Sutcliffe Efficiency (NSE∗) loss
function proposed by Kratzert et al. (2019) to avoid an imbalance during training due to the higher influence of catchments
with a higher runoff generation.

## 6.2 Setup conceptual models

The lumped conceptual model used in CAMELS-DE is called "simple hydrological model (SHM)" and is a variant of the
well-known HBV (Hydrologiska Byråns Vattenbalansavdelning; Bergström and Forsman, 1973) model. A detailed
description of the model architecture and setup can be found in the studies by Ehret et al. (2020) and Acuña et al. (2024).
SHM uses mean precipitation and potential evapotranspiration ($E_{pot}$; mm d$^{-1}$) as inputs. The $E_{pot}$ is calculated using the
temperature-based Hargreaves formula, detailed by Adam et al. (2006) and based on earlier work by Droogers and Allen
(2002), as explained and cited in Clerc-Schwarzenbach et al. (2024). This variant of Hargreaves formula resulted in the
lowest mass balance error in most catchments with respect to other methods (e.g. Penman, Priestly Taylor) to estimate
evapotranspiration and was additionally chosen due to its low data requirements, enabling the utilisation of HYRAS
precipitation and temperature data to generate the $E_{pot}$ time series with a limited number of assumptions. The $E_{pot}$ time series
are included in CAMELS-DE (Tab. 2) for the entire time period of 70 years. In terms of model training, the SHM was
trained individually for each basin using the NSE as a loss function, employing the Differential Evolution Adaptive
Metropolis (DREAM; Vrugt, 2016) algorithm as implemented in the SPOTPY (SPOTting model parameters using a
ready-made PYthon package, Houska et al., 2015) library. In contrast to the LSTM the SHM model is mass conserving and
hence more sensitive to errors in the catchment delineation that can lead to mass balance errors (see section 3). The
difference between the SHM and the LSTM performance can be seen as an indicator either for a strong human influence or
for an imprecise catchment delineation as the LSTM can create mass.

## 6.3 Results LSTM and SHM model

In this section, we focus our analysis on the LSTM and SHM model in catchments where at least 20 % of the daily data is
available during the 30-year training period and 10 % during the testing period, covering a total of 1384 catchments. The
median performance of the LSTM, as quantified by the NSE during the testing period, is 0.84 across 1384 catchments. Of
these, 91 catchments have an NSE lower than 0.5 (6.6 % of all catchments), out of which 27 have a negative NSE (1.95 % of
all catchments). For the 91 catchments with NSE below 0.5, most streamflow time series exhibit a low Pearson correlation
with daily precipitation (< 0.1) and these catchments are often considerably affected by the construction and/or operation of
dams or flood control structures (human influences attributes). Therefore, model performance of the LSTM network can be



used to identify catchments that are subject to considerable uncertainties, either due to measurement inaccuracies or
significant human influences.

Fig. 5a illustrates the performance of the LSTM model across various federal states, with relatively consistent results across
the board except for the federal states of Brandenburg (DE4) and Saxony-Anhalt (DEE). In Brandenburg, lowland
catchments characterised by sandy soils, considerable groundwater impacts, abundance of natural lakes and human
constructed weirs, canals and cross-connections between streams most likely yield a distinctly lower model performance
compared to the rest of the German federal states. Besides the federal state of Brandenburg and Saxony-Anhalt the analysis
of the LSTMs simulations reveals no clear correlation between the model performance and the topographic attributes (e.g.,
area), climatic attributes (e.g., long-term mean precipitation), or hydrological attributes (e.g., long-term mean flow).

The performance of the conceptual model is with a median NSE of 0.71 lower than that of the LSTM (Fig. 5b). In 188
catchments (13.6 %) the conceptual model shows a performance below a NSE of 0.5 and in 43 (3.1 %) a performance below
a NSE of 0. The spatial patterns of performance measured by the NSE are interestingly consistent between the LSTM and the
conceptual model. In other words, catchments where the LSTM performs well are typically also accurately represented by
the conceptual model, and vice versa, as illustrated in Fig. 5e. Catchments in cases the conceptual model significantly
underperforms compared to the LSTM  are almost invariably strongly influenced by human-made structures such as dams or
weirs, or they are located in areas with uncertain catchment delineation. We propose that the conceptual model, which
conserves mass and uses time-invariant parameters, struggles to adapt to dynamic changes in catchment function caused by
human activities that result in inaccuracies in water flow and storage due to structures like dams, weirs or due to irrigation or
pumping. A hypothesis that requires further testing in the few catchments where this is the case.







**Figure 5:** Panel (a) shows boxplots visualising the distribution of the NSE of the LSTM network (blue) and the conceptual model (orange) for each federal state in Germany for the testing period. Panel (**b**) shows a cumulative plot of the NSE for the general comparison of the LSTM model and the conceptual model. Panel (c) shows the NSE values of the LSTM for all 1555 gauging stations in Germany, while panel (c) shows the same for the NSE values of the conceptual model. Panel (e) shows the difference between the NSE values of the LSTM and the conceptual model for all gauging stations in Germany, borders of Germany: © GeoBasis-DE / BKG (VG250, 2023)

## 7 Code availability, reproducibility and extensions

The processing of CAMELS-DE is structured in a modular manner to enhance the clarity and reproducibility of the processing pipeline. The CAMELS-DE processing pipeline was published separately with more details and permalinks to the



released repository versions that represent the code state that was used to process and compile CAMELS-DE (Dolich, 2024).
For each component of CAMELS-DE, a distinct GitHub repository was established. Within each repository, a dedicated
Docker container was developed to process specific input datasets (e.g. HYRAS, GLO-30 DEM). Containerization is
particularly well-suited for this project as it ensures that each component of the data processing pipeline runs consistently
across different computing environments. This containerization simplifies dependency management, enhances
reproducibility, and facilitates the deployment and version control of each processing module. Fig. 6 illustrates the
architecture of the processing pipeline, where each blue block represents an individual GitHub repository equipped with a
Docker container that processes the yellow input data to produce the green output data. All repositories are uniformly
structured, and the accompanying documentation provides detailed descriptions of each repository, guidelines for building
and running the Docker containers, including the necessary folder mounts, and instructions for accessing the required input
data. In the initial phase of the CAMELS-DE data processing pipeline, raw discharge and water level data, along with station
metadata provided by the federal states, are processed and harmonised. Subsequently, MERIT-Hydro catchment boundaries
are delineated for each station, a pivotal step since all further datasets depend extensively on these catchment boundaries.
Meteorological time series data for these catchments are then processed to compute statistics such as area mean and median.
Following this, attributes such as soil properties, hydrogeology, land cover, topography, and human influences are derived for
each catchment (see Table 2). In the final stage, all derived data are integrated and formatted according to the established
structure of the CAMELS-DE dataset, mirroring the organisational schema of CAMELS-GB or CAMELS-CH.



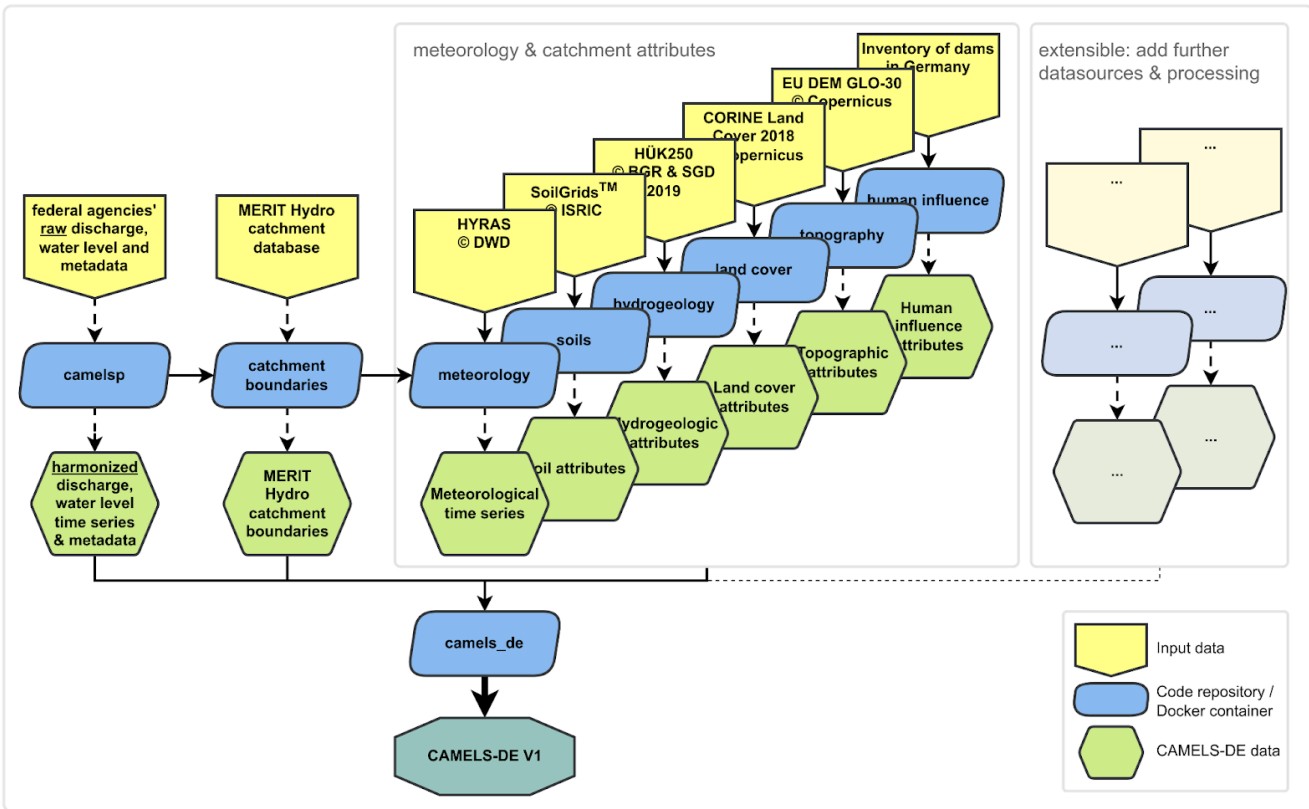


**Figure 6:** Diagram of the CAMELS-DE data processing pipeline. Starting with raw discharge and metadata harmonisation, it proceeds to derive MERIT-Hydro catchment boundaries. Subsequent processing includes meteorological data extraction and aggregation followed by the extraction of various catchment attributes. In the final step, all extracted data sources are integrated in the structured CAMELS-DE dataset, consistent with CAMELS-GB or CAMELS-CH (Dolich, 2024).

The modular design of the CAMELS-DE processing pipeline enhances its traceability, comprehensibility, and reproducibility, differing significantly from a monolithic code approach that compiles the entire dataset into a single repository. This structure not only facilitates the extension of the pipeline to incorporate additional data sources, especially further catchment attributes, without the need to re-run or rewrite the entire system but also allows for the adaptation of processing or aggregation methods and the seamless release of updated versions of the CAMELS-DE dataset. The publicly available Docker containers and the code within them serve not only as a comprehensive guide to understanding the data processing methods used in CAMELS-DE but also provide a foundation for further data processing using the catchment geometries included in the dataset. We encourage researchers to enrich CAMELS-DE with additional data sources and explore ways to enhance the baseline model results. Such contributions are invaluable for continuous improvements and expansions of the CAMELS-DE dataset, reflecting our commitment to advancing hydrological research and applications through reproducible science.



## 8 Data availability

CAMELS-DE and is freely available at https://doi.org/10.5281/zenodo.12733968 (Dolich et al., 2024), accompanied by a comprehensive data description. The code to reproduce CAMELS-DE can be found at https://doi.org/10.5281/zenodo.12760336 (Dolich, 2024).

## 9 Conclusions

CAMELS-DE is a significant step forward in hydrological research for Germany and beyond, offering a comprehensive dataset that spans 1555 catchments with hydro-meteorological daily time series from 1951 to 2020. CAMELS-DE includes detailed catchment delineations and properties, such as reservoir data, land-use, soils, and hydrogeology, which are all vital to analyse and describe the local and regional hydrology of Germany. Furthermore, CAMELS-DE includes simulations from a regionally trained LSTM and locally trained conceptual model that can be used either to fill gaps in discharge data in case of good model performance or act as baseline models for the development and testing of new hydrological models. Due to the length of the provided time series of up to 70 years CAMELS-DE opens up new opportunities for investigating long-term hydrological trends or conducting large-sample studies across diverse catchments, including a large number of catchments smaller than 100 km². The dataset's modular design, achieved through the containerization of each processing component, ensures that the data processing is traceable, comprehensible, and reproducible. This approach makes it easier to extend the dataset by incorporating new data sources, adapting processing methods, and releasing updated versions without the need to re-run the entire pipeline. While CAMELS-DE serves as a useful benchmark for large sample hydrology, we invite the scientific community to enrich it with additional data sources and improved methods. In conclusion, CAMELS-DE aims to support a broad range of hydrological research and applications, to foster better understanding and management of water resources in Germany and beyond and to contribute to future global hydrological studies.

**Author contribution:** RL and MS initiated the CAMELS-DE project. AD prepared and processed data, created most figures and wrote together with RL most of the manuscript. All other authors suggested improvements and made additions to the manuscript, as well as provided data and expertise for specific topics.

**Competing interests:** At least one of the (co-)authors is a member of the editorial board of Earth System Science Data or Hydrology and Earth System Sciences.

**Acknowledgment**: We thank the various German institutions for providing observation-based data and sharing their expertise. We are grateful to the Volkswagen Foundation for funding the "CAMELS-DE" project within the framework of





the project "Invigorating Hydrological Science and Teaching: Merging Key Legacies with New Concepts and Paradigms"
(ViTamins). We also extend our thanks to NVDI4 Earth, particularly Jörg Seegert, for their support and suggestions.

**Table 2.:** Catchment-specific static attributes available in CAMELS-DE

| Attribute class | Attribute name | Description | Unit | Data source |
|---|---|---|---|---|
| Location and topography | gauge_id | catchment identifier based on the NUTS classification as described in section 5.1 e.g. DE110000, DE110010, … | – | Federal state agencies (see section 2) |
| | provider_id | official gauging station ID assigned by the federal states | – | |
| | gauge_name | gauging station name | | |
| | water_body_name | water body name | – | |
| | federal_state | federal state in which the measuring station is located | | |
| | gauge_lon | gauging station longitude (EPSG:4326) | ° | |
| | gauge_lat | gauging station latitude (EPSG:4326) | ° | |
| | gauge_easting | gauging station easting (EPSG:3035) | m | |
| | gauge_northing | gauging station northing (EPSG:3035) | m | |
| | gauge_elev_metadata | gauging station elevation as given by the federal states | m.a.s.l. | |
| | area_metadata | catchment area as given by the federal states | km² | |
| | gauge_elev | gauging station elevation derived from the GLO-30 DEM | m a.s.l. | Copernicus GLO-30 DEM (EU-DEM, 2022) |
| | area | catchment area derived from the MERIT Hydro catchment | km² | |
| | elev_mean | mean elevation in the catchment based on the MERIT Hydro geometry | m a.s.l. | |
| | elev_min | minimum elevation within catchment | m a.s.l. | |
| | elev_5 | 5th percentile elevation within catchment | m a.s.l. | |
| | elev_50 | median elevation within catchment | m a.s.l. | |
| | elev_95 | 95th percentile elevation within catchment | m a.s.l. | |



| | elev_max | maximum elevation within catchment | m a.s.l. | |
|---|---|---|---|---|
| Climate | p_mean | mean daily precipitation | mm d$^{-1}$ | German Weather Service HYRAS (DWD-HYRAS, 2024) |
| | p_seasonality | seasonality and timing of precipitation (estimated using sine curves to represent the annual temperature and precipitation cycles, positive (negative) values indicate that precipitation peaks in summer (winter), and values close to zero indicate uniform precipitation throughout the year). | – | |
| | frac_snow | fraction of precipitation falling as snow, i.e. while mean air temperature is < 0° C | – | |
| | high_prec_freq | frequency of high-precipitation days ($\geq$ 5 times mean daily precipitation) | d yr$^{-1}$ | |
| | high_prec_dur | mean duration of high-precipitation events (number of consecutive days $\geq$ 5 times mean daily precipitation) | d | |
| | high_prec_timing | season during which most high-precipitation days occur, e.g. 'jja' for summer. If two seasons register the same number of events a value of NA is given. | season | |
| | low_prec_freq | frequency of dry days (< 1 mm d$^{-1}$) | d yr$^{-1}$ | |
| | low_prec_dur | mean duration of dry periods (number of consecutive days < 1 mm d$^{-1}$ mean daily precipitation) | d | |
| | low_prec_timing | season during which most dry season days occur, e.g. 'son' for autumn. If two seasons register the same number of events a value of NA is given. | season | |
| Hydrology | q_mean | mean daily specific discharge | mm d$^{-1}$ | Federal state agencies (see section 3.1) and German Weather Service HYRAS (DWD-HYRAS, 2024) |
| | runoff_ratio | runoff ratio (ratio of mean daily discharge to mean daily precipitation) | – | |
| | flow_period_start | first date for which daily streamflow data is available | – | |
| | flow_period_end | last day for which daily streamflow data is available | – | |
| | flow_perc_complete | percentage of days for which streamflow data is available from Jan 1951–31 Dec 2020 | % | |
| | slope_fdc | slope of the flow duration curve (between the log-transformed 33rd and 66th stream flow percentiles, see Coxon et al. (2020) | – | |
| | hfd_mean | mean half-flow date (number of days since 1. Oct at which the cumulative dis charge | d | |



| | | | | |
|---|---|---|---|---|
| | | reaches half of the annual discharge) | | |
| | Q5 | 5 % flow quantile (low flow) | mm d$^{-1}$ | |
| | Q95 | 95 % flow quantile (high flow) | mm d$^{-1}$ | |
| | high_q_freq | frequency of high-flow days ((> 9 times the median daily flow) | d yr$^{-1}$ | |
| | high_q_dur | mean duration of high-flow events (number of consecutive days > 9 times the median daily flow) | d | |
| | low_q_freq | frequency of low-flow days (< 0.2 times the mean daily flow) | d yr$^{-1}$ | |
| | low_q_dur | mean duration of low-flow events (number of consecutive days < 0.2 times the mean daily flow) | d | |
| | zero_q_freq | fraction of days with zero stream flow | – | |
| Land cover | artificial_surfaces_perc | areal coverage of artificial surfaces | % | CORINE Land Cover 2018 (CLC, 2018) |
| | agricultural_areas_perc | areal coverage of agricultural areas | % | |
| | forests_and_seminatural_areas_perc | areal coverage of forests and semi-natural areas | % | |
| | wetlands_perc | areal coverage of wetlands | % | |
| | water_bodies_perc | areal coverage of water bodies | % | |
| Soil | clay_0_30cm_mean clay_30_100cm_mean clay_100_200cm_mean | weight percent of clay particles (< 0.002 mm) in the fine earth fraction at depths 0 - 30 cm, 30 - 100 cm and 100 - 200 cm | wt. % | SoilGrids250m (Poggio et al., 2021) |
| | silt_0_30cm_mean silt_30_100cm_mean silt_100_200cm_mean | weight percent of silt particles (≥ 0.002 mm and ≤ 0.05/0.063 mm) in the fine earth fraction at depths 0 - 30 cm, 30 - 100 cm and 100 - 200 cm | wt. % | |
| | sand_0_30cm_mean sand_30_100cm_mean sand_100_200cm_mean | weight percent of sand particles (> 0.05/0.063 mm) at depths 0 - 30 cm, 30 - 100 cm and 100 - 200 cm | wt. % | |
| | coarse_fragments_0_30cm_mean coarse_fragments_30_100cm_mean coarse_fragments_100_200cm_mean | volumetric fraction of coarse fragments (> 2 mm) at depths 0 - 30 cm, 30 - 100 cm and 100 - 200 cm | vol % | |
| | soil_organic_carbon_0_30cm_mean soil_organic_carbon_30_100cm_mean soil_organic_carbon_100_200cm_mean | soil organic carbon content in the fine earth fraction at depths 0 - 30 cm, 30 - 100 cm and 100 - 200 cm | g kg$^{-1}$ | |



| | bulk_density_0_30cm_mean<br>bulk_density_30_100cm_mean<br>bulk_density_100_200cm_mean | bulk density of the fine earth fraction at depths 0 - 30 cm, 30 - 100 cm and 100 - 200 cm | kg dm$^{-3}$ | |
|---|---|---|---|---|
| Hydrogeology | aquitard_perc<br>aquifer_perc<br>aquifer_aquitard_mixed_perc | areal coverage of aquifer media type classes | % | HÜK250 © BGR & SGD (Staatlichen Geologischen Dienste) 2019 (HGM, 2019) |
| | kf_very_high_perc (>1E-2 m s$^{-1}$)<br>kf_high_perc (>1E-3 – 1E-2 m s$^{-1}$)<br>kf_medium_perc (>1E-4 – 1E-3 m s$^{-1}$)<br>kf_moderate_perc ( (>1E-5 – 1E-4 m s$^{-1}$)<br>kf_low_perc (>1E-7 – 1E-5 m s$^{-1}$)<br>kf_very_low_perc (>1E-9 - 1E-7 m s$^{-1}$)<br>kf_extremely_low_perc (<1E-9 m s$^{-1}$)<br>kf_very_high_to_high_perc (>1E-3 m s$^{-1}$)<br>kf_medium_to_moderate_perc (>1E-5 – 1E-3 m s$^{-1}$)<br>kf_low_to_extremely_low_perc (<1E-5 m s$^{-1}$)<br>kf_highly_variable_perc<br>kf_moderate_to_low_perc (>1E-6 – 1E-4 m s$^{-1}$) | areal coverage of permeability classes | % | |
| | cavity_fissure_perc<br>cavity_pores_perc<br>cavity_fissure_karst_perc<br>cavity_fissure_pores_perc | areal coverage of cavity type classes | % | |
| | consolidation_solid_rock_perc<br>consolidation_unconsolidated_rock_perc | areal coverage of consolidation classes | % | |
| | rocktype_sediment_perc<br>rocktype_metamorphite_perc<br>rocktype_magmatite_perc | areal coverage of rock type classes | % | |
| | geochemical_rocktype_silicate_perc<br>geochemical_rocktype_silicate_carbonatic_perc<br>geochemical_rocktype_carbonatic_perc<br>geochemical_rocktype_sulfatic_perc<br>geochemical_rocktype_silicate_organic_components_perc<br>geochemical_rocktype_anthropogenically_modified_through_filling_perc<br>geochemical_rocktype_sulfatic_halitic_perc<br>geochemical_rocktype_halitic_per | areal coverage of geochemical rock type classes | % | |



| | c | | | |
|---|---|---|---|---|
| | waterbody_perc | areal coverage of water body areas according to hydrogeological map | % | |
| | no_data_perc | percentage of areas with missing data | % | |
| Human influence | dams_names | names of all dams located in the catchment | – | Inventory of dams in Germany (Speckhann et al., 2021) |
| | dams_river_names | names of the rivers where the dams are located | – | |
| | dams_num | number of dams located in the catchment | – | |
| | dams_year_first | year when the first dam entered operation | – | |
| | dams_year_last | year when the last dam entered operation | – | |
| | dams_total_lake_area | total area of all dam lakes at full capacity | km$^2$ | |
| | dams_total_lake_volume | total volume of all dam lakes at full capacity | Mio m³ | |
| | dams_purposes | purposes of all the dams in the catchment | – | |
| Hydrological Simulations | training_perc_complete | percentage of observed specific discharge values in the training period (1970-10-01 – 1999-12-31) that are not NaN | % | Regional LSTM model, conceptual model (see section 6, https://github.com/KIT-HYD/Hy2DL/tree/v1.1, last access: 24 July 2024) |
| | validation_perc_complete | percentage of observed specific discharge values in the validation period (1965-10-01 – 1970-09-30) that are not NaN | % | |
| | testing_perc_complete | percentage of observed specific discharge values in the testing period (2001-10-01 – 2020-12-31) that are not NaN | % | |
| | NSE_lstm | Nash-Sutcliffe model efficiency coefficient of the LSTM in the testing period | – | |
| | NSE_conceptual | Nash-Sutcliffe model efficiency coefficient of the conceptual model in the testing period | – | |

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
