# Peer review of "CAMELS-DE: hydro-meteorological time series and attributes for"

_Earth System Science Data, 2024_

## Author Response (AR1)

Dear Editor,

First of all, thank you very much for handling our manuscript. We have updated our figures and improved the manuscript in accordance with the reviewers' suggestions. This includes renaming the SHM to HBV including the additional references and minor enhancements to the description of our meteorological forcing data.

Additionally, we have updated our dataset to include catchments from the federal state of Saarland, which were not part of the initial dataset version as we were awaiting permission. This inludes that we had to update the title slighlty.

Furthermore, since the dataset has been available to the community, we received feedback that allowed us to make further improvements to both the manuscript and the dataset during the review process. These updates include addressing feedback regarding negative discharge values, which can occur in northern Germany due to tidal influences, uploading the trained LSTM and HBV model weights (parameters) with the dataset, and correcting 0 values in one catchment where NaN values were expected.

Thank you once again for your support.

Best regards,
Ralf Loritz (on behalf of the co-authors)

---

## Author Response (AR2)

Dear Conrad Jackisch, thank you for handling our manuscript.

CJ: "*Please note that you have to include the correct DOI and citation as last sentence in the abstract. https://www.earth-system-science-data.net/submission.html#manuscriptcomposition This needs to be added. Moreover, you indicated to publish the final dataset with GFZ or PANGAEA as a curated repository. I see the advantages with Zenodo, but I would like to encourage you to check if you can have a nice combination of the bleeding edge development in GitHub/Zenodo and the well-checked freezes at GFZ?*"

RL: Thank you for bringing this to our attention. We have updated the abstract to include the correct DOI and citation as per the submission guidelines. Regarding the publication of the final dataset, after careful consideration and discussion within our team, we have decided to keep the dataset on Zenodo. The link provided in the manuscript directs to version 1.0 of the dataset, which corresponds precisely to the data described in our paper (this Version is not going to change). Zenodo offers us the flexibility to release updated versions of the CAMELS dataset in the future, all within the same repository, ensuring continuity and ease of access for users. While we acknowledge the advantages of curated repositories like GFZ or PANGAEA for well-checked data freezes, we believe that Zenodo better suits our needs for ongoing development and version control. We hope you understand our decision and are confident that this approach will best serve the community.

"*Fig 3a is certainly wild with a streetmap as background. I do not like it and do not see why the basemap is presented if there is no information about the exemplary catchment is provided nor shining through the data overlay. A terrain shade plus rivers would be much mor appropriate. You might consider to update this.*"

RL: We changed the figure according to your comments.

CJ: "*Fig 5a&b: Why did you opt to turn the axis from y to x in 5b? I would consider the distribution plot as some sort of marginal distribution which can use a joint y axis instead? This could also resolve the odd placement of the legend somewhere in the middle? I suggest to plot 5b as KDEs and to find a better place for the legend. You might consider to update this.*"

RL: Thank you for your suggestions regarding Figures 5a and 5b. We understand your concerns about the axis orientation and the placement of the legend. However, we have decided to retain the current format of Figure 5b to maintain consistency with the presentation style commonly used in machine learning papers published in HESS. This consistency facilitates easier comparison with existing literature and helps readers familiar with this format to interpret the results more effectively

Again thank you for handling our MS. Sincerely

Ralf Loritz